# Breakpoint:
# Scalable evaluation of system-level reasoning in LLM code agents

**Kaivalya Hariharan**[*]
MIT
kaivu@mit.edu

**Uzay Girit**[*]
MIT
zef@mit.edu

**Atticus Wang**
MIT
atticusw@mit.edu

**Jacob Andreas**
MIT
jda@mit.edu

## Abstract

Benchmarks for large language models (LLMs) have predominantly assessed short-horizon, localized reasoning. Existing long-horizon suites (e.g. SWE-lancer) rely on manually curated issues, so expanding or tuning difficulty demands expensive human effort and evaluations quickly saturate. However, many real-world tasks, such as software engineering or scientific research, require agents to rapidly comprehend and manipulate novel, complex structures dynamically; evaluating these capabilities requires the ability to construct large and varied sets of problems for agents to solve. We introduce Breakpoint, a benchmarking methodology that automatically generates code-repair tasks by adversarially corrupting functions within real-world software repositories. Breakpoint systematically controls task difficulty along two different dimensions: local reasoning (characterized by code complexity metrics such as cyclomatic complexity) and system-level reasoning (characterized by call-graph centrality and the number of simultaneously corrupted interdependent functions). In experiments across more than 900 generated tasks we demonstrate that Breakpoint's methodology can scale to arbitrary difficulty, with state-of-the-art models' success rates ranging from 55% on the easiest tasks down to 0% on the hardest. We analyze how static parameters control task difficulty, characterize how improvements in models and inference-time budgets affect local versus system-level reasoning, and evaluate the strategies models use to gather information and iterate on solutions, demonstrating Breakpoint's effectiveness as a comprehensive evaluation suite for understanding agent behavior and capabilities.

## 1 Introduction

Language models (LMs) have rapidly advanced in recent years, achieving remarkable performance in mathematical problem-solving (Glazer et al., 2024) and programming tasks (Miserendino et al., 2025). Increasingly, LMs are used not just for fixed instruction-following tasks, but as autonomous agents interacting with (and influencing) the outside world through code and digital interfaces (Liu et al., 2024).

Success in these long-horizon coding tasks should ideally test and disambiguate two separate abilities:

- **Localized reasoning** — fixing a self-contained bug or implementing a single, clearly-defined feature.
- **System-level reasoning** — diagnosing and coordinating changes across complex, interconnected codebases.

While these are both required for effective software development, being a good local reasoner does not imply directly being a good systems reasoner. However, existing software-

---

[*]Equal contribution; first-author order was randomized.

**1. Select a target function or component**

```python
def _verify_hmac(params):
    ...
    signature = _sign_hmac(hash_algorithm_name, sig_base_str,
                           client_secret, resource_owner_secret)
    match = safe_string_equals(signature, request.signature)
    return match
```

**2. Corrupt Function with different transformations:**

Function Removal

```python
def _verify_hmac(params):
    pass
```

LLM Generated Corruption

```python
def _verify_hmac(params):
    # setting up encyption components
    ...
    if client_secret is not None and resource_owner_secret is not None:
        signature = _sign_hmac(hash_algorithm_name, sig_base_str,
            resource_owner_secret, client_secret)
    else:
        signature = _sign_hmac(hash_algorithm_name, sig_base_str,
            client_secret, resource_owner_secret)
    ...
```

**3. Run agent to explore and fix the codebase**

list/read files, search code

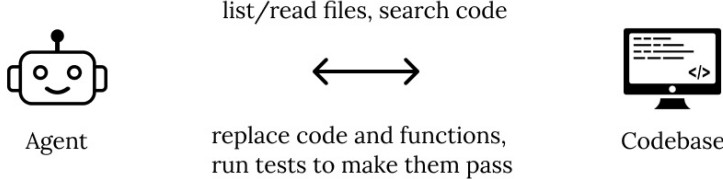

Agent   replace code and functions,   Codebase
        run tests to make them pass

Figure 1: Breakpoint Inverse Problem Methodology.

engineering benchmarks, such as SWEBench (Jimenez et al., 2024), predominantly emphasize localized reasoning by curating tasks derived from individual GitHub issues, or deliberately selecting problems that are isolated and straightforwardly evaluatable. Even benchmarks designed to test longer-horizon interactions, such as SWELancer (Miserendino et al., 2025), rely heavily on manual human curation and thus cannot easily scale in difficulty or scope. Consequently, despite impressive performance in isolated contexts, it has remained unclear whether today's models truly *understand* and can reliably manipulate the complex systems they interact with.

This paper describes **Breakpoint**, a benchmark (and benchmark construction framework) designed to explicitly evaluate and distinguish these capabilities through automatically generated *code-repair* tasks (Figure 1). Breakpoint adversarially corrupts or removes core, well-tested architectural components within real GitHub repositories (see Section C.1 for selection criteria), and challenges agents to restore functionality solely by using feedback from induced test failures. Because the full call graph is observable, Breakpoint is able to systematically control and measure task difficulty along two distinct dimensions: **Local complexity**, characterized by the complexity of corrupted functions, and **system-level complexity**, characterized by both by call-graph centrality and the number of simultaneously corrupted, interdependent functions.

These metrics enable fine-grained evaluation of whether improvements in models or increased computational budgets primarily affect localized reasoning, system-level reasoning, or both.

Across 930 Breakpoint tasks, we find that current state-of-the-art models achieve meaningful success (up to 55%) on simpler single-function repairs but struggle (0%) on more challenging task settings. Our results show that task difficulty can be systematically controlled and decomposed into clear local and system-level reasoning components, enabling fine-grained analysis of model capabilities. Additionally, providing models with greater inference-time budgets substantially improves their performance, particularly on tasks requiring deeper systems understanding. Finally, detailed analysis on agent tool usage reveals that models engage in substantially different information gathering behaviors.

## 2   Related Work

**"Real-World" Code Evaluations**   Several recent benchmarks evaluate models on real-world software to assess coding ability. **SWE-bench** (Jimenez et al., 2024) tasks models with resolving actual GitHub issues within popular open-source repositories. **SWELancer** (Miserendino et al., 2025) benchmarks models against real-world freelance tasks from Upwork, incorporating tasks ranging from bug fixes to high-level project management. Breakpoint similarly targets coding ability on real world codebases, and uses existing test suites to validate correctness. However, while our inverse problem methodology requires no labeled problem data, these benchmarks rely on extensive human labeling or existing collaborative traces. This allows Breakpoint to easily scale to very challenging problems.

**Inverse Problems**   R2E similarly lifts the curation restriction by synthesizing an "oracle harness": models are tasked with recreating function bodies given context, and then the correctness is verified by equality on LLM-generated test cases (Jain et al., 2024). This method also does not require labeled curation, but R2E selects for tasks that can easily be separated from the codebase into their own environment. Breakpoint focuses on precisely those that are embedded deeply into the codebase as a better means of understanding long horizon ability.

**General Agent Benchmarks and Long-Horizon Reasoning.**   Recent work has begun to probe how well large language models retain goals and intermediate state over hundreds of steps. The *METR Long-Task Report* shows steady scaling in the maximum length of tasks that frontier models can complete (Kwa et al., 2025). **AgentBench** aggregates eight interactive environments and highlights persisting weaknesses in memory and decision consistency (Liu et al., 2024). Other domain-specific sets include MARPLE (multi-modal crime reconstruction) and FrontierMath (advanced proofs spanning pages of chain-of-thought) (Jin et al., 2024; Glazer et al., 2024). Breakpoint has a similar focus on long horizon ability, but Breakpoint's methodology leverages the existing structure of codebases to generate long horizon problems grounded in the intrinsic structure of the system, without requiring manual creation or curation of long horizon tasks.

## 3   Breakpoint Methodology

In Breakpoint, each task starts from a real, test-passing repository that we view as a tuple $(F, T, G)$: a set of functions/classes $F$, a test suite $T$, and a call graph $G$. We then adversarially corrupt one or more functions in $F$ to create a "broken" repo that flips previously passing tests (see Fig. 1). The agent gets a small toolbox to navigate and edit code plus a fixed budget of tool uses and test runs (edits persist within a trajectory), and its job is to repair the corrupted definitions so the repository returns to an all-tests-passing state within the budget. Scoring is binary: success if all tests pass at the end of the run, failure otherwise (see Appendix B for interaction details).

### 3.1 Task Modes

Breakpoint supports three corruption modes and two interaction settings that together probe complementary aspects of reasoning.

**Corruption modes.**

1. **Function Deletion.** The corrupted function $\tilde{f}_c^{del}$ retains only the original signature and documentation; the implementation is removed entirely.

2. **Adversarial Corruption.** A separate language model introduces subtle, syntactically valid changes to a function, yielding $\tilde{f}_c^{adv}$. These edits are designed to be hard to detect and repair; see Appendix E.

3. **Multifunction Corruption.** A set of structurally related functions—chosen to be within a small chain distance in the call graph—are simultaneously adversarially corrupted, creating interdependent bugs that induce systemic failures (Fig. 3a).

**Interaction settings.**

- REMOVE **mode.** Used for function deletion tasks. The task description identifies the deleted target, and the agent may only modify that function. This isolates the repair step and tests targeted local reasoning (see Appendix B for tool details).

- DISCOVERY **mode.** Used for adversarial and multifunction corruptions. The task description does not reveal which function(s) are corrupted; the agent only sees failing tests and must first *localize* the fault(s) and then repair them. To avoid reward hacking, trajectories that pass tests without modifying the correct function(s) are scored as failures. (Edits persist across the trajectory; see Appendix B.)

Together, these settings let us disentangle capabilities: REMOVE emphasizes information gathering and targeted local repair when the fault location is known, while DISCOVERY stresses diagnosis and system-level reasoning to isolate subtle anomalies before repair.

### 3.2 Complexity Dimensions

Throughout our work we quantify the following function properties, allowing us to scale difficulty and analyze model performance on different kinds of problems.

**Evaluating System-Level Reasoning.** As a proxy to select functions or components that will require system-level reasoning, we measure the **harmonic centrality** $H(f)$ of each function $f \in F$ over the call graph $G$. Harmonic centrality reflects how "close," on average, the rest of the system is to $f$.

Intuitively, functions with higher harmonic centrality significantly influence many other parts of the system, either directly or through short indirect call chains, thereby requiring deeper systemic understanding for effective repair. Empirically, harmonic centrality is correlated with other centrality measures, including PageRank, harmonic, and pure eigenvector centrality.

**Evaluating Local Reasoning.** To measure intrinsic complexity within individual functions, we employ **code-line complexity**, a simple baseline that proxies a function's complexity based on the number of lines of code it contains.

While local reasoning encompasses more than just code length, this metric serves as a reasonable proxy because longer functions typically require understanding more logic branches, variables, and algorithmic steps within a bounded scope. Successfully repairing complex functions demonstrates at least some capacity for localized reasoning—the ability to comprehend and fix intricate logic without needing to trace through the broader system.

Empirically, other complexity measures, such as Halstead complexity, lines-of-code, and nesting-depth metrics, are highly correlated with code-line count.

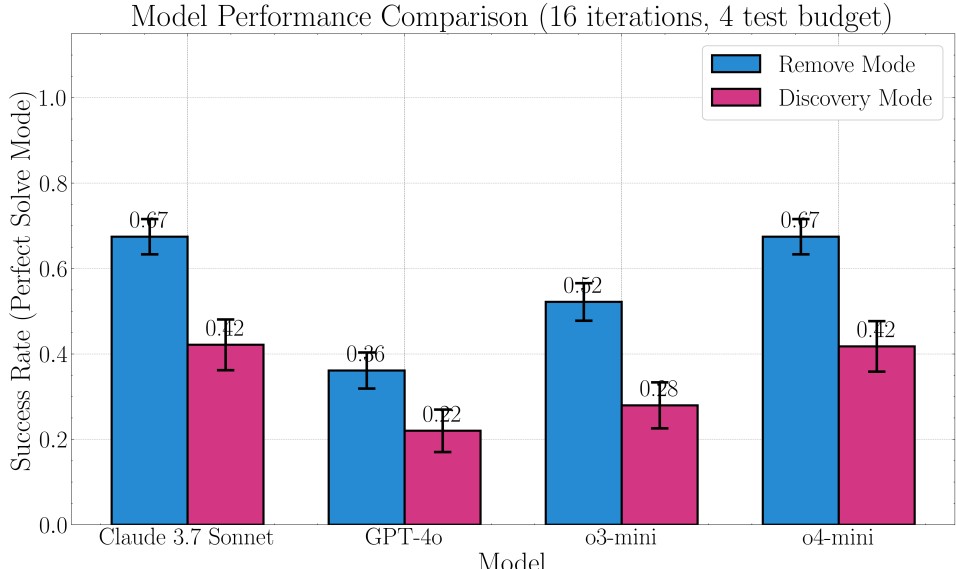

Figure 2: **Breakpoint effectively assesses and differentiates models based on their ability to solve long horizon coding tasks.** Comparison of models across targeted removal and corruption in discovery mode

## 4 Experiments and Results

Our experimental evaluation addresses three core questions:

1. **Controlling Task Difficulty:** To what degree can we predict and control task difficulty through adjustments of static parameters in *Breakpoint*?

2. **Model Scaling and Differentiating Capability Improvements:** When models either improve through specialized reasoning abilities or are given increased inference-time computation (more tool-use iterations), do they primarily improve in localized problem-solving or in deeper system-level reasoning?

3. **Understanding agent behavior:** What information-gathering behaviors and iteration strategies characterize different models?

By default, models are allocated four times as many tool-use calls as they have submission attempts, typically configured at 16 tool-use iterations and 4 submissions unless otherwise specified. To investigate scaling effects, we also specifically evaluate models o4-mini and gpt-4o across varying inference-time computation budgets (4/1, 8/2, 16/4, and 32/8 tool-use calls/submissions respectively). Detailed prompts and a full list of available tools are provided in the appendix.

### 4.1 Finding a scalable, static measure of problem difficulty in code environments

#### 4.1.1 Breakpoint Problems are Extremely Challenging for Frontier Models

We can leverage Breakpoint's inverse problem methodology to source a set of highly challenging problems. We find that we are able to scale both the difficulty of the local reasoning and the system-level reasoning required to solve these problems.

1. **Multifunction Corruption significantly increases Discovery Mode difficulty.** We induce multiple simultaneous corruptions of interrelated functions (which we define as functions that are at most 4 steps apart on the call graph): in doing so, we generate challenging tasks that require strong diagnostic capabilities. We find that performance sharply deteriorates as the number of corruptions increases,

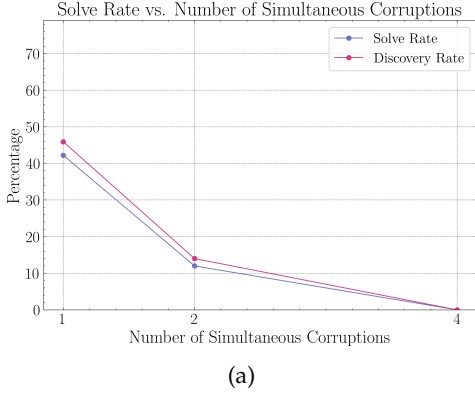
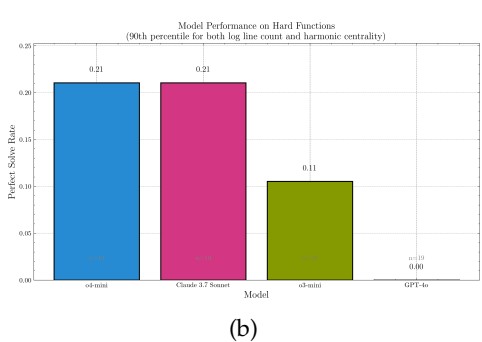

(a)             (b)

Figure 3: **Left:** Performance of o4-mini (with 32 tool calls and 8 test iterations allowed) sharply declines as the number of simultaneous, semantically related corruptions increases, ultimately reaching a 0% success rate. **Right:** To construct our "hard" set, we select tasks at or above the 90th percentile in both complexity (code-line count) and centrality (harmonic centrality in the call graph). This rigorous selection criterion ensures that only the most challenging, system-level problems are included.

       ultimately rendering even state-of-the-art models unable to solve any scenarios with 4 simultaneous corruptions (see Figure 3a).

2. **Targeted Selection of High Complexity-Centrality tasks substantially increases difficulty in Mode.** Selecting functions high in both structural centrality and functional complexity creates difficult tests of both systemic and localized reasoning. Selecting problems that are at least 90th percentile in complexity and centrality measures reduces overall performance by nearly 40 % per model.

### 4.1.2 *Centrality & Complexity Jointly Explain Difficulty in* REMOVE *Mode*

We find that we can decompose difficulty in REMOVE tasks along two dimensions: (i) local code complexity and (ii) function centrality. This allows us to scale the difficulty of REMOVE tasks and automatically generate hard questions.

To disambiguate these terms, we perform a joint logistic-regression analysis on *z*-scored *code-line count* (a proxy for cyclomatic complexity) and *harmonic centrality* (our default centrality metric). While these metrics show some correlation (see Appendix F), they capture distinct aspects of difficulty: individual models using only code-line count achieve McFadden's $R^2$ = 0.216 (AIC: 4163.1), while models using only harmonic centrality achieve $R^2$ = 0.183 (AIC: 4340.9). The combined model improves to $R^2$ = 0.221 (AIC: 4141.4), demonstrating that both features contribute complementary information. We find that *both* predictors carry significant negative coefficients (two–sided Wald tests, $p < 10^{-4}$ for each), confirming that success probability falls off with increasing complexity *and* centrality. Effect sizes differ in magnitude: every +1SD in code lines suppresses the odds of success by $\approx 55\%$, whereas a +1SD in harmonic centrality suppresses them by $\approx 20\%$.

Figure 4 visualises these trends across the full task space, whereas Figure 12 shows more granular per-model difficulty scaling, making it clear that the monotone difficulty gradient in these axes is robust to model choice and persists even for the best-performing systems.

### 4.1.3 *Discovery Mode Tests Practical "Needle-in-the-Haystack" Reasoning*

Given the clear decomposition of difficulty provided by static complexity and centrality metrics in the REMOVE (targeted) mode (see Section 4.1.2), one might expect similar patterns to emerge in the DISCOVERY mode. Surprisingly, we find that static metrics do not provide a clear decomposition in this more open-ended setting.

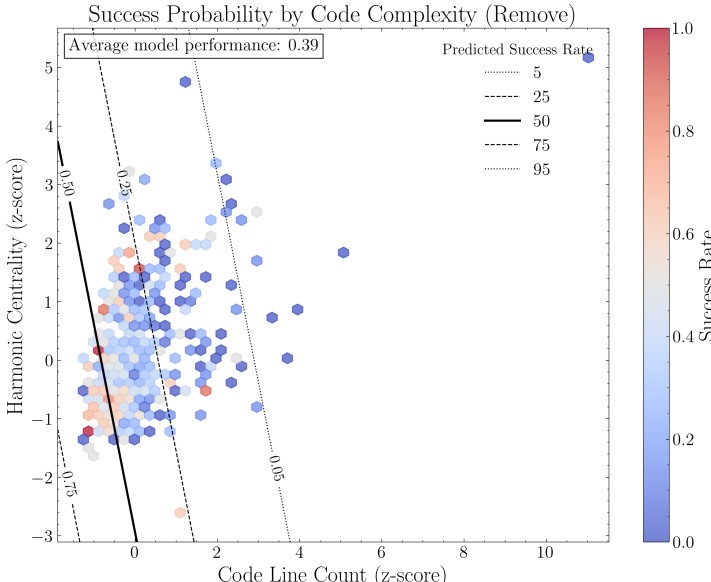

Figure 4: **Joint influence of complexity and centrality on task success in Remove mode.** Each hexagon aggregates tasks lying in a 2-D bin of *z*-scored code-line count (x-axis) and harmonic centrality (y-axis); color encodes the empirical success rate (red = easy, blue = hard). The solid black line is the zero-logit decision boundary from the joint regression above; dotted and dashed lines mark the 75th and 95th percentile difficulty contours predicted by that model.

In practice, models face two distinct challenges in Discovery mode: identifying the corrupted function(s) (diagnostic step) and subsequently repairing the identified issues (repair step). Empirically, we observe a strong but not absolute correlation between diagnostic success and overall task success (see Figure 10) in the Appendix. Specifically, successfully pinpointing the corrupted function substantially improves the odds of solving the task—conditional success rates range between 59% and 92%—highlighting diagnosis as a critical bottleneck. However, a non-trivial gap remains, particularly for models that do less reasoning, suggesting that even after correct diagnosis, the repair step presents meaningful difficulties.

## 4.2 Model Scaling and Differentiating Capability Improvements

In this section, we study precisely how agent performance varies for different models and inference time compute budgets. We tease apart two different sources of improvement: (i) **reasoning improvement** (o4-mini vs. gpt-4o) and (ii) **run-time interaction budgets** (more tool–test cycles). For each source we ask which difficulty axis—*local complexity* or *systems difficulty*—is most explanatory. To do so, we use two complementary analysis.

- **Logistic difficulty curves** (as in Fig. 12 and 11) plot the *success probability* of a *single* model as a smooth function of a difficulty metric. Parallel curves imply uniform difficulty scaling across all models, while diverging slopes imply axis-specific gains.

- **Mann–Whitney U tests** compare the *distributions of difficulty values among the tasks that each model* actually solved. Because the test is rank-based, it is insensitive to the uniform-shift scenario: if model B simply solves "everything model A does, plus a few tougher items everywhere," the rank distributions stay similar.

We therefore use *both*: curves tell us *how* probability changes within a model, while Mann–Whitney tells us *whether* the extra wins cluster along one axis.

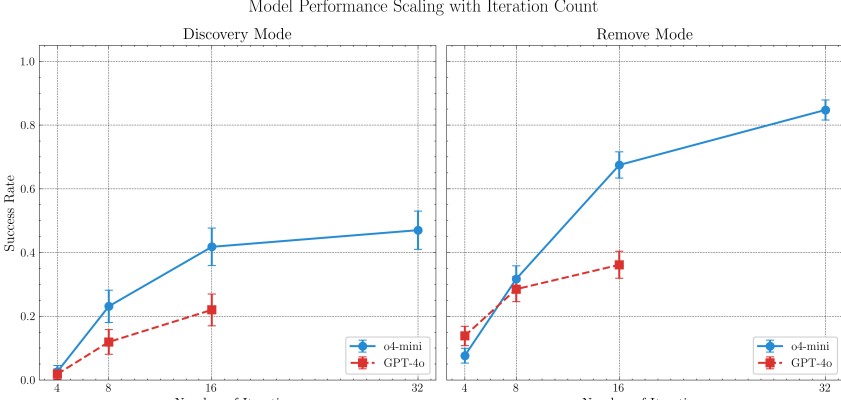

Figure 5: **Model performance improves when scaling both iteration and test feedback**. We measure performance as we scale max iterations and test budget simultaneously in both task modes. We observe that in remove mode o4-mini is continuously improving whereas gpt-4o performance saturates. Iteration is less beneficial for both models in discovery mode.

### 4.2.1   *Model performance improves with environmental interaction*

We study how the model performance scales with increased interaction with the environment and submission attempts. We find that the model performance increases with environmental interaction.

In Figure 5 we measure performance as we scale max iterations and test submission budget (see Appendix B for agent details) simultaneously so that the model is able to iterate and submit more, and we find that the models are able to solve more questions when given more attempts and access to information, but that the improvements plateau.

In Figure 9 we retroactively measure model performance on successive repair attempts in the same trajectory. Models are clearly able to to effectively diagnose failures and then gather information and improve to fix them. We find that most of the gains happen after the first test iteration.

### 4.2.2   *Reasoning model improvements differentially benefit localized reasoning*

We study the centrality and complexity distributions of tasks solved by `gpt-4o` and `o4-mini` to understand how reasoning improvements manifest across different difficulty dimensions. A Mann–Whitney test on *code-lines* yields $p = 0.019$ ($r = 0.21$), whereas the same test on *harmonic centrality* is non-significant ($p = 0.47$).

This implies post-training reasoning gains differentially benefit solving longer, more complex functions: in other words, the jump in performance from reasoning is better explained by the model improving at localized reasoning than at system-level reasoning. Figure 14 in the Appendix visualizes these distributions through violin plots, clearly showing the shift toward higher complexity functions for `o4-mini` while centrality distributions remain similar.

### 4.2.3   *Both system-level reasoning and localized reasoning scales with inference time environmental interaction*

We compared the distribution of solved problems for a low iteration run of o4-mini (8 tool calls and 2 submissions) vs high iterations (32 tool calls and 8 submissions). The Mann-Whitney U test revealed that increased iterations resulted in statistically significant improvements for both centrality and complexity, but the effect size (Cohen's d) is much stronger for centrality (0.49) than complexity (0.15).

This indicates that additional tool use and iterations significantly enhance the model's performance in resolving issues within more centrally connected parts of the system. The gains are explained by the model being able to do both more localized, but especially more system-level reasoning via iteration. Figure 11 in the Appendix highlights this improvement evenly across difficulty dimensions, while the full distributions are visualized in Figure 13 in the Appendix.

## 5 Discussion

**Using inverse problems to generate difficult yet evaluatable tasks.** The difficulty of reliably benchmarking sophisticated reasoning capabilities often arises from an inherent tension between complexity and evaluability. Traditionally, hard reasoning tasks are created in isolation.

Inverse problems resolve this tension by utilizing pre-existing, complex structures—such as extensive software codebases. By starting with an already functioning and verifiable system, and deliberately introducing adversarial corruptions, we generate challenging tasks that explicitly demand deep, systemic comprehension. Because the original, correct state of the system is known, performance can be precisely measured through objective metrics like existing test suites. This structure enables robust evaluation at scales and complexities beyond what single-question tasks typically allow, potentially even scaling to tasks beyond single-human generative capability, since they are generated by extensive collaborative processes over long periods of time.

**Evaluating system-level reasoning and testing models for real-world integration.** System-level reasoning is what allows humans to quickly understand companies, research fields, codebases, etc. to then modify and improve them in precise ways. Real-world software systems contain this kind of systemic complexity—where a single modification can cascade through interconnected components, potentially causing unpredictable systemic failures. By specifically targeting this capability, Breakpoint highlights the critical importance of reasoning about interconnected dependencies, structural coherence, and broader implications of local changes.

**Decomposing long horizon model reasoning for a science of agents** Current methodologies for evaluating language model performance, such as measuring loss scaling, offer limited insight into the high-level capabilities models actually possess. Our approach and metric analysis propose decomposition of model abilities, and offer steps towards a better understanding of the inputs and outputs of resource scalings.

Understanding precisely how these factors impact reasoning difficulty can inform theoretical frameworks and future model training strategies, moving beyond superficial performance measures towards deeper diagnostic tools capable of pinpointing specific cognitive or representational gaps in language models.

This precise analysis along with the problem framing allow us to study the agentic behavior of different model in long horizon settings, and on various types of problems - answering what kind of tools they use, how they respond to failed attempts, and how much they seek information. We are excited about future work going deeper into a potential science of agent behavior.

**Inspect integration** To facilitate broader use, we ported Breakpoint to Inspect (AI Security Institute)—a popular open source evaluation standard—thereby enabling drop-in execution and head-to-head comparison of state-of-the-art software-engineering (SWE) agents on our tasks.

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

# A Preliminaries

Breakpoint tasks require agents to solve coding problems within pre-existing software repositories. Formally, we can represent a repository $R$ as a tuple $(F, T, G)$, where:

- $F = \{f_1, f_2, \ldots, f_n\}$ is a finite set of functions or classes, with each $f_i$ defined by signature, implementation, and documentation.
- $T = \{t_1, t_2, \ldots, t_m\}$ is a set of tests, each $t_j : R \mapsto \{0, 1\}$, verifying the correctness of the repository's functionality (1 indicating passing).
- $G = (V, E)$ is the call graph, a directed graph where vertices $V = F$ represent functions or classes, and edges $E \subseteq F \times F$ represent their calls between one another, with an edge $(f_a, f_b)$ indicating $f_a$ calling $f_b$.

A **benchmark task instance** is then defined by adversarially modifying a function or set of functions $f_c \in F$, creating a corrupted version $\tilde{f}_c$, and generating a corrupted repository:

$$\tilde{R} = (F \setminus \{f_c\} \cup \{\tilde{f}_c\}, T, G).$$

See Fig. 1 for an example such task instance.

The goal for a model $M$ is to produce a repaired version of the function $\hat{f}_c$ or the larger set of functions, such that the updated repository $\hat{R}$ restores the original functionality as validated by tests:

$$\hat{R} = (F \setminus \{f_c\} \cup \{\hat{f}_c\}, T, G).$$

The model is allowed a fixed number of tool and test interactions to submit patches in a standard agent setup (see more info in Appendix B).

# B Agent details

The LM agent is provided with a set of tools to interact with the codebase. They always include:

- list_directory: List files and directories at a specified path.
- search_code: Search for a text or regex pattern across all .py files.
- read_file: Read a file, if it's too large the file functions are listed for a future read_function call.
- list_file_functions: Return a list of top-level function and class definitions in a file.
- read_function: Read the contents of a specified function or class from a file.

In targeted mode (the modes are described in **??**), the agent has the additional tool:

- `submit_attempt`: Submit an attempt at fixing the targeted function or class.

In discovery mode, where the target function is unknown the above tool is replaced with:

- `replace_function`: Submit new code for a target function or class by replacing its definition and the body.

In discovery mode the modifications are persistent throughout a trajectory.

We run the model with a tool-use budget `max_tool_uses` to limit the number of times the agent can interact with the codebase. We also set a separate `max_attempts` to the total number of times either `submit_attempt` or `replace_function` can be invoked; this is the maximum number of times the agent can test out the code it writes and get error reports.

## C   Additional Experimental Details

### C.1   Data selection

We scrape codebases from GitHub to source problem instances. We choose repositories based on:

- **Test Coverage:** We select for inverse problems that make at least 5 previously passing tests fail, and we select for codebases where all tests initially pass.
- **Test Efficiency:** Test suites must execute within 60 seconds of wall-clock time, facilitating efficient benchmarking.
- **Popularity and Quality:** Repositories are required to have at least 1000 GitHub stars to ensure some standard of quality and testing.

Currently, Breakpoint focuses on Python repositories for consistency and ease of automation, though the method generalizes broadly.

## D   Studying how agents interact with the environment

We analyze how models interact with the breakpoint code environment, and in particular tool usage behavior.

### D.0.1   *Different models have different tool usages*

We find that the best models use more tools in general and are more information seeking, with o4-mini being the most proactive about tool usage (see Figure 6), although Claude 3.7 uses less tools and is as performant.

We analyze different models' strategies via their tool calling behavior, and measure their usage of different tools in Figure 7. We classify tools into those that read off information from the environment, and those that attempt to modify it to solve the task. We measure the average use of these two types in Figure 6.

More successful models (1) gather more information before submitting, (2) use targeted search tools effectively, and (3) learn efficiently from test feedback - suggesting that progress in system-level reasoning may come from better environmental exploration strategies and ability to iterate as much as from improved reasoning capabilities.

### D.1   Human Evaluation of Task Difficulty

To better contextualize the difficulty of Breakpoint tasks, we conducted a preliminary human evaluation with the authors solving a small set of benchmark tasks. Specifically, four problems from each of the REMOVE and **discovery** modes were manually attempted. All eight tasks were successfully solved, with completion times ranging from 4 to 30

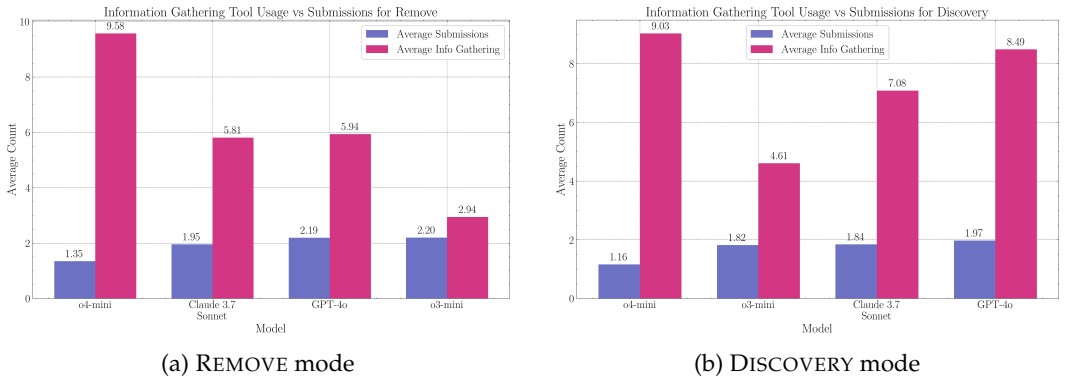

(a) REMOVE mode         (b) DISCOVERY mode

Figure 6: Average information-gathering tool calls (y-axis) versus code-submission attempts (x-axis) for each model. `o4-mini` is the most information-seeking model in both task modes.

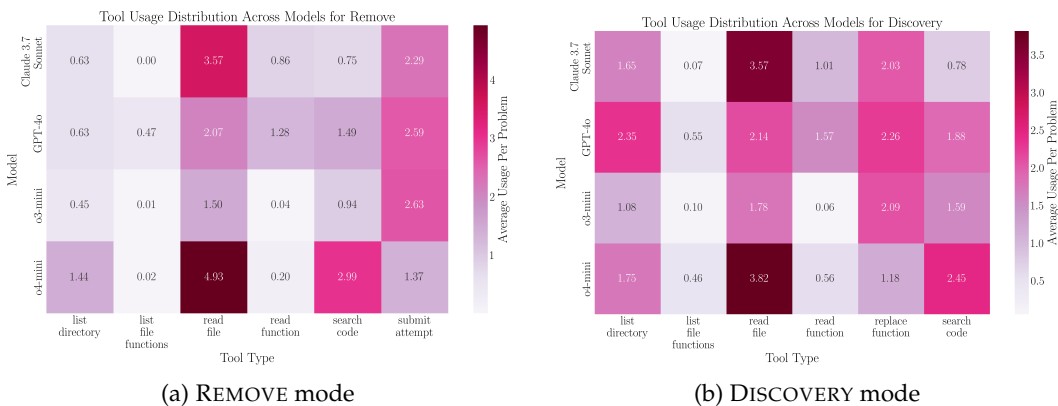

(a) REMOVE mode         (b) DISCOVERY mode

Figure 7: Distribution of tool usage by model. color intensity indicates the average number of calls per tool type, per problem.

minutes per task. These results suggest that Breakpoint tasks—at least at lower corruption complexities—are within reasonable human solvability.

Qualitatively, we found that single-corruption tasks in the **discovery** mode generally felt easier to solve compared to analogous tasks in the REMOVE mode. Informal experiments further indicated that even moderate assistance from language models substantially accelerates human solving times, particularly in REMOVE mode, highlighting potential synergies between human and model reasoning.

We also qualitatively assessed the difficulty of the hardest Breakpoint problems by sampling one hard set problem for both remove and discovery. Based on our human evaluation, we found that the tasks are quite challenging and would require several hours of human time to understand and fix the codebase, although they are solvable. It qualitatively seems that models are better at quickly processing large amounts of information for coding but they struggle to solve the hardest problems that require longer time horizons, whereas humans are much slower but more reliably solve the problem.

To facilitate future systematic comparisons, our benchmark release includes explicit support for human evaluations, providing users with interfaces to efficiently explore, diagnose, and repair tasks directly within the codebase environment.

# E   Model Corruptions

To generate corruptions, we prompt OpenAI's o4-mini model with the following prompt template (below).

It can iterate 5 times on the test feedback before it submits a final attempt for the corruption. This corruption is then stored and applied before the solving model tries to do the problem.

```
You are an expert Python programmer tasked with introducing a bug in a function located in {function_path}.

Your goal is to modify the following function so that some of the test cases fail.
DO NOT INDICATE WITH COMMENTS
OR GIVE ANY CLUE AS TO WHERE THE BUG IS,
OR WHERE YOU MODIFIED THE FUNCTION,
as your modified function will be a test for another agent.

Use the following heuristic: if another agent were to get this codebase without knowing where
the corruption is, and were to see the new test errors,
it would be hard to find the function where it was caused and
it would be hard to then,
given the function, fix the problem with it.

Your goal is to make it subtle, not necessarily to make all the original tests fail,
although at least 2 tests should fail.

Inserting some esoteric add-on or string that will show up in tests
and can easily be found via searching the code
would also not work well with this heuristic.

For example, your corruption should involve the actual behavior of the function,
a bad idea for this heuristic would just be a syntax error
or making the function empty or whatever.

A good idea using this heuristic would be to see how
the function interacts with the rest of the code and identify an area
where you can subtly damage the code.

Motivate your corruptions by looking at how the tests work and what can make them fail.

Do not change the function definition, just copy it over.

Keep asking yourself: If I saw this error in this codebase,
would I easily fix it? If yes, iterate and do better.

Function to corrupt:
{func_code}

To help guide you to what is being tested, here are some relevant tests that you can try to make fail.

{test_examples}

You have access a submit_attempt tool that allows you to
submit attempted patches for the function,
and then get test feedback (max {self.test_budget} times).
Reflect on the errors and if they are subtle/hard to debug.
You can call tools at most {self.max_iterations} times.

Remember that the only way to submit your code is via the submit_attempt tool.
```

```
Again, DO NOT INDICATE WITH COMMENTS OR GIVE ANY CLUE AS TO WHERE THE BUG IS,
OR WHERE YOU MODIFIED THE FUNCTION.
```

# F Complexity and Centrality Metrics

In this appendix, we provide definitions of the complexity and centrality metrics used throughout our analysis. These metrics characterize both the local complexity of individual functions and their structural importance within the broader codebase. We then analyze the correlations between these metrics to understand their relationships and implications for debugging difficulty.

## F.1 Complexity Metrics

We employ several established software complexity metrics to quantify the local difficulty of individual functions:

**Lines of Code (LOC)** The simplest measure of function complexity, defined as the total number of coding (not comment or docstring) lines in a function's implementation.

**Cyclomatic Complexity** Cyclomatic complexity $M$ measures the number of linearly independent paths through a function's control flow graph:

$$M = E - N + 2P$$

where $E$ is the number of edges, $N$ is the number of nodes, and $P$ is the number of connected components (typically $P = 1$ for a single function). In practice, we compute this as $M = 1 +$ the number of decision points (conditionals, loops, exception handlers).

**Halstead Metrics** Halstead metrics treat programs as collections of operators and operands:

- **Halstead Difficulty**: $D = \frac{\eta_1}{2} \cdot \frac{N_2}{\eta_2}$, where $\eta_1$ and $\eta_2$ are the numbers of distinct operators and operands respectively, and $N_2$ is the total count of operands.
- **Halstead Volume**: $V = N \cdot \log_2(\eta)$, where $N = N_1 + N_2$ is the total number of operators and operands, and $\eta = \eta_1 + \eta_2$ is the vocabulary size.

## F.2 Centrality Metrics

To capture the structural importance of functions within the codebase, we compute several graph centrality measures on the function call graph $G$:

**PageRank Centrality** Adapted from **?**, PageRank measures a function's importance based on the importance of its callers. For a function $f$, its PageRank score is computed iteratively:

$$PR(f) = \frac{1 - d}{|V|} + d \sum_{g \in \text{callers}(f)} \frac{PR(g)}{|\text{out}(g)|}$$

where $d = 0.85$ is the damping factor, callers$(f)$ denotes functions that call $f$, and $|\text{out}(g)|$ is the out-degree of function $g$. High PageRank indicates functions that are called by many important functions, typically representing core utilities or fundamental components.

**Harmonic Centrality** Following **?**, harmonic centrality measures a function's ability to efficiently reach other functions in the call graph:

$$HC(f) = \frac{1}{|V| - 1} \sum_{g \in V \setminus \{f\}} \frac{1}{d(f, g)}$$

where $d(f, g)$ is the shortest path distance from $f$ to $g$, with $\frac{1}{d(f,g)} = 0$ if $g$ is unreachable from $f$. This metric identifies high-level orchestrator functions that control many parts of the codebase.

**Distance Discount Centrality**   We introduce a decay-based centrality measure that captures diminishing influence over distance:

$$DD(f) = \sum_{g \in \text{reachable}(f)} \alpha^{d(f,g)}$$

where $\alpha = 0.5$ is the decay factor. This metric assigns exponentially decreasing weights to functions based on their distance: direct callees contribute 0.5, distance-2 functions contribute 0.25, and so forth.

**Betweenness Centrality**   Based on **?**, betweenness centrality measures how often a function appears on shortest paths between other functions:

$$BC(f) = \sum_{s \neq t \neq f \in V} \frac{\sigma_{st}(f)}{\sigma_{st}}$$

where $\sigma_{st}$ is the number of shortest paths from $s$ to $t$, and $\sigma_{st}(f)$ is the number of those paths passing through $f$. High betweenness indicates "bridge" functions that connect different modules of the codebase.

**Degree Centralities**   We also compute basic degree measures:

- **In-degree**: $d_{in}(f) = |\{g \in V : (g, f) \in E\}|$ (number of callers)
- **Out-degree**: $d_{out}(f) = |\{g \in V : (f, g) \in E\}|$ (number of callees)
- **Total degree**: $d(f) = d_{in}(f) + d_{out}(f)$

### F.3   Correlation Analysis

Figure 8 presents the correlation matrix between all computed metrics. We observe several patterns:

**Strong Complexity Correlations**   Complexity metrics exhibit high mutual correlations ($r > 0.75$):

- LOC and cyclomatic complexity ($r = 0.751$) correlate strongly, as longer functions typically contain more decision points.
- Cyclomatic complexity and Halstead volume ($r = 0.887$) show the strongest correlation, indicating that control flow complexity closely tracks the total number of operations.
- Cyclomatic complexity and Halstead difficulty ($r = 0.767$) also correlate highly, suggesting that complex control flow involves more intricate operator usage patterns.

**Centrality Metric Relationships**   Among centrality measures, we observe:

- Harmonic and distance discount centralities correlate moderately ($r = 0.726$), as both capture outgoing influence with different decay functions.
- PageRank shows low correlation with harmonic and distance discount centralities, reflecting their opposite directionality (incoming vs. outgoing calls).
- Degree metrics naturally correlate, with total degree and in-degree showing $r = 0.897$.

**Success Rate Correlations**   The relationship between metrics and debugging success reveals:

- Complexity metrics negatively correlate with success rates, confirming that more complex functions are harder to debug.
- Outgoing centrality measures (harmonic, distance discount) show stronger relationships with success than incoming measures, indicating that functions controlling many others present greater debugging challenges.

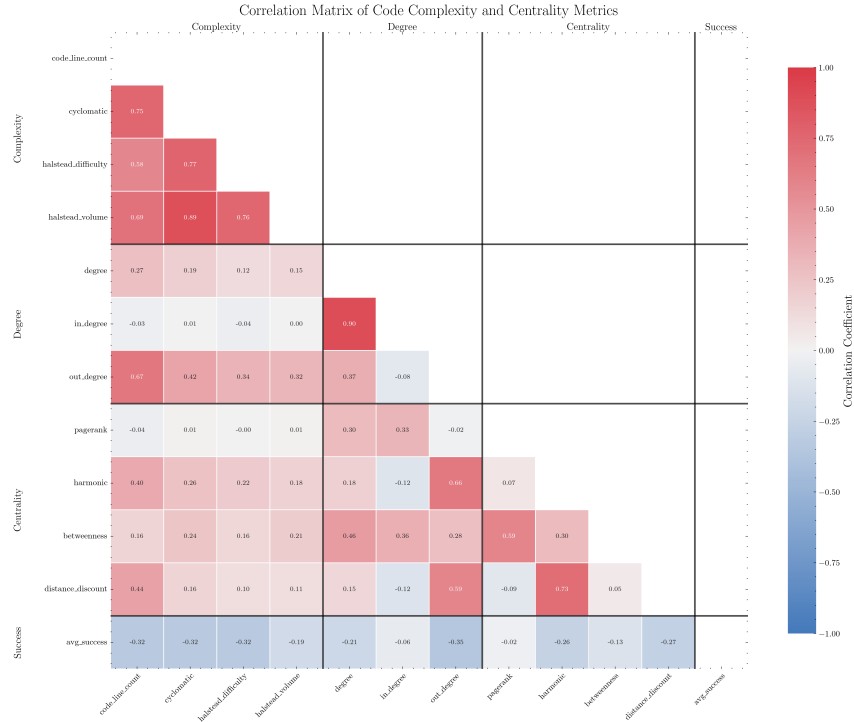

Figure 8: Correlation matrix of code complexity and centrality metrics. Metrics are grouped into four categories: Complexity (lines of code, cyclomatic complexity, Halstead metrics), Degree (in/out/total degree), Centrality (PageRank, harmonic, betweenness, distance discount), and Success (average success rate across models).

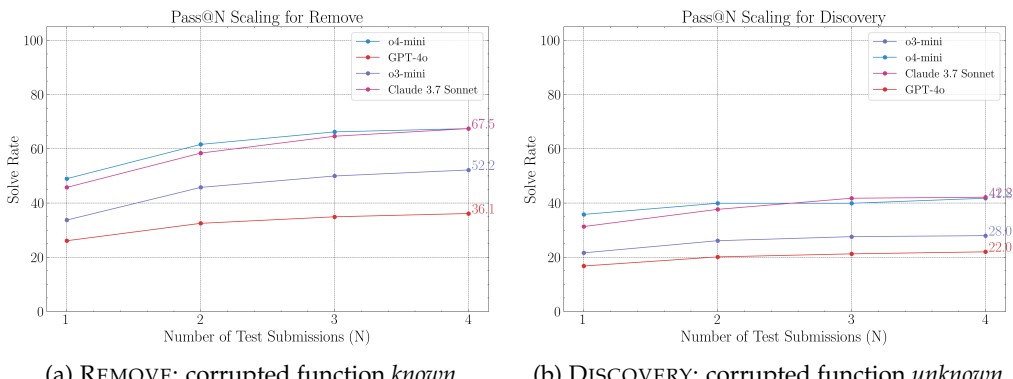

(a) REMOVE: corrupted function *known*        (b) DISCOVERY: corrupted function *unknown*

Figure 9: **Test feedback helps—but almost entirely on the first revision.** We measure the cumulative success rate after the $n^{\text{th}}$ submitted patch—we compute these curves *within the same trajectory*: after each submitted patch we ask, "If we had stopped here, what fraction of tasks would already be solved?" In REMOVE mode (left) every model gains a visible boost on the **second** submission, then plateaus. In DISCOVERY mode (right) even the 2nd submission barely improves the score, suggesting diminishing returns beyond the first attempts.

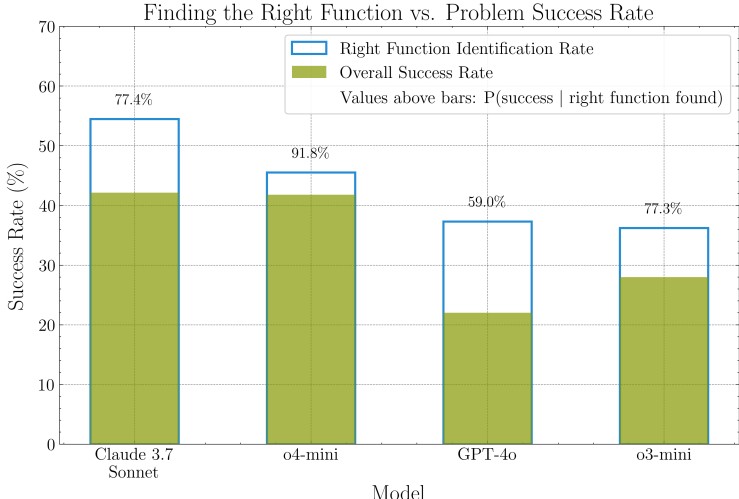

Figure 10: **Discovery mode success depends heavily on diagnostic accuracy.** Blue bars indicate the proportion of tasks where models correctly identify at least one corrupted function. Green bars show the proportion of overall successful repairs. Conditional success rates (reported above bars) significantly exceed unconditional rates, emphasizing that locating the corruption is a critical challenge. Nevertheless, considerable variation in conditional success highlights meaningful differences among models' local repair capabilities.

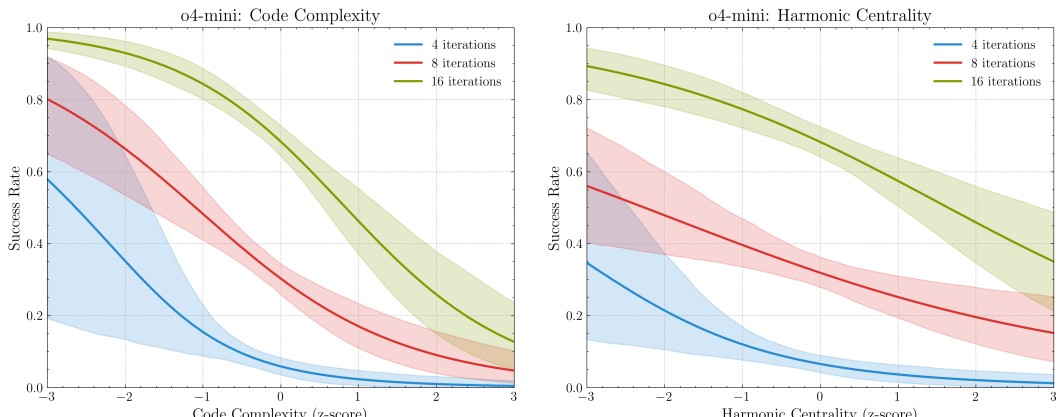

Figure 11: **Logistic curve fit to the solve probability in terms of centrality and complexity as iterations are varied**. Increased iteration helps across the board on both dimensions.

## G  Model Performance Scaling

## H  Violin Plots for Differentiating Capabilities

To provide additional visual insight into how different sources of improvement affect model performance, we present violin plots showing the distribution of task difficulty metrics (code-line complexity and harmonic centrality) among problems successfully solved by each model or configuration. These plots complement the statistical analyses presented in the main text by visualizing the full distribution of solved tasks along each difficulty dimension.

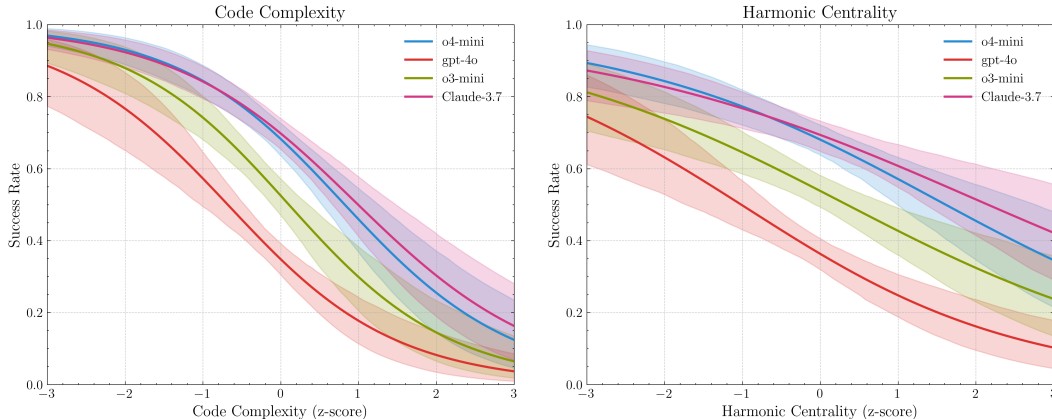

Figure 12: **Per-model logistic difficulty curves.** Left: predicted success probability versus code-line count; Right: versus harmonic centrality, each at 16 tool uses/4 submissions. Shaded bands give 95% CIs from bootstrapped fits. Although the absolute success levels differ (o4-mini and Claude 3.7 dominate; GPT-4o lags), the downward trend with each predictor is consistent across models, reinforcing the additive difficulty-decomposition found in the pooled analysis.

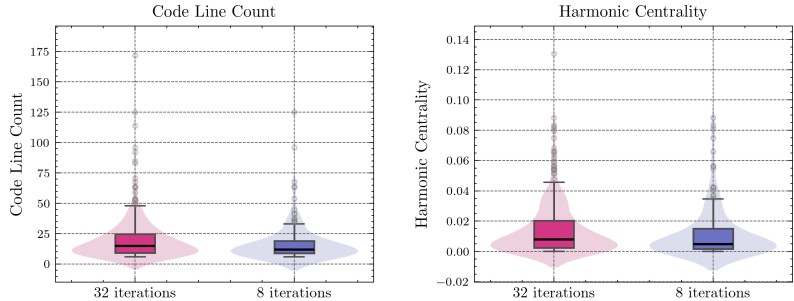

Figure 13: **Increased interations improve both localized and especially system-level reasoning.** We measure the distribution of harmonic centrality (**right**) and code line count (**left**) of problems solved as we vary the amount of interactions the models are given with the environment. Given more iterations o4-mini is better at central and complex problems, with a strong effect size for centrality as measured by the Mann-Whitney U Test.

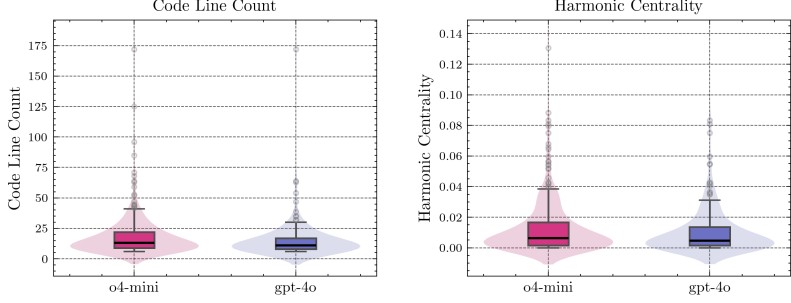

Figure 14: **Reasoning gains are differentially concentrated in localized reasoning**. The colored regions highlight the number of functions solved by each model as we vary code line count (**left**) and harmonic centrality (**right**). The points at the peak of the distribution indicate the hardest functions solved by each model. The Mann Whitney U test shows that complexity but not centrality differentiates between the two models, suggesting o4-mini's performance is explained by stronger localized reasoning.

