# OpenReview forum: "Breakpoint: Stress-testing systems-level reasoning in LLM agents"
_colmweb.org/COLM/2025/Conference — COLM 2025_

### Official Review · Reviewer_zGgD · 2025-05-12

**Rating:** 6
**Confidence:** 4
**Ethics Flag:** 1

**Summary:**

This works presents a new benchmark Breakpoint to test the real-world software development capabilities of LLMs. Breakpoint is created by systematically corrupting existing software respoistories and asks the LLM-powered agents to fix the corresponding corrupted code. The evaluation done in the paper shows that Breakpoint is extremely challenging for current models.

**Questions To Authors:**

Please address the concerns I had listed in the Reasons to Reject section.

**Reasons To Accept:**

- The problem of benchmarking current LLM-based software agents is an important problem to work on
- The authors release a complete toolkit which allows users to adopt Breakpoint for their own repo as well as for future work to build on top of it

**Reasons To Reject:**

Comparions with other benchmarks:
- Compared to other benchmarks, I would argue that Breakpoint seems to be the most unrealistics
- The authors simply modifies the current functions or remove them to create artifical problems
- Its hard to see if these problems are indeed similar to the real-world software developement problems that are part of the other benchmarks

Limited Novelty:
- The idea of the work is "corrupt" the existings repo by either modifying the fucntions or removing them entirely to create a new benchmark
- This is akind to similar ideas in traditional software engineering such as inserting synthetic bugs or mutation testing
- Breakpoint does not seem to extend any of those prior ideas and there is no additional interesting insights to be gained even looking at sections 4.4 and 5

Limited Evaluation:
- While the authors did a good job conducting some evaluation according to the "problem difficulty"
- The authors only used on agent/technique to test on the benchmark.
- For a benchmarking work, I would expect the authors to select multiple existing tools to see how they perform
- As such the claim that the "best-performing models fail entirely on our most difficult tasks" is a bit misleading as not all existing techniques have been evaluated

Minor:
- The authors seem to have missed another similar benchmarking work: https://openreview.net/forum?id=MMwaQEVsAg (I am not the author of) which constructs a repo from scratch (but the idea can be done for partial repos)

---

> ### Author Response · Authors · 2025-06-03
> **In response to helpful feedback**
>
> Thank you for your feedback. Please see our responses to the helpful points raised in your review below:
>
> > “unrealistic” — synthetic vs real bugs
>
> We believe our tasks are representative of important skills in software engineering.
>
> In remove mode, the model is given a function and needs to implement it. This is very similar to classic SWE workflows where given some large codebase a developer needs to implement a well scoped out functionality.
>
> For our discovery mode, this is similar to a bug or backwards compatibility being introduced in a codebase, and then a developer having to diagnose where it lies and fix it. In fact, our evaluation shows much of the difficulty in remove mode lies in finding where the bug is.
>
> Our results remain interesting beyond the scope of evaluating SWE ability. We want to understand the ability of AI agents to dynamically explore and control a complex system, of which codebases are a first class example. This general ability is very broad and important for real world integration, in and outside of software engineering.
>
> Our methodology also allows us to select and create difficult tasks that can scale in difficulty beyond human level. In our qualitative human testing we found that our hardest sets were very challenging for experienced programmers, suggesting our evaluation is able to sample and select for very hard problems.
>
> We [link a figure reflecting how having synthetic bugs allows us to scale difficulty in this way](https://paste.pics/c96ff0a74bc10d7d16ac834a2e72200b). In our n-corruption mode where we corrupt related function components, we are able to create a hard set where the model can solve 0% of the problems. We also select on centrality/complexity to filter tasks in remove mode, creating a much harder set for current models (right). This task creation and selection on difficulty is possible because our problems are synthetic.
>
> > "Limited novelty — akin to mutation testing"
>
> We appreciate the reviewer's attention to this important distinction. While Breakpoint shares surface similarities with mutation testing, it fundamentally differs in purpose, methodology, and insights gained. We respectfully disagree about the claim on the limited novelty of the paper.
>
> Our goal in this paper is not to develop better mutation testing tools for evaluating software artifacts; instead, it's to apply automated code transformation methods (including, but not limited to, mutation) to understand if and how an LLM‑powered agent can reason about and repair a fault whose effects cascade through a real, multi‑module codebase.
>
> Our benchmarking methodology is innovative for the following reasons:
>
> - **Graph‑aware corruption.** We select targets by the joint distribution of complexity and centrality, then introduce semantic edits (multi‑line changes or deletions) that propagate across module boundaries—faults that ordinary mutation pipelines rarely create. As far as we know, this is a novel way of sourcing hard problems, that allows us to explicitly target systems level reasoning.
> - **Creation of discovery and targeted modes.** By separating localisation (discovery) from patch synthesis (targeted), Breakpoint tests for a variety of the subskills involved in SWE tasks.
> - **Predictive metrics for scalable generation of difficult tasks**. We show that two simple graph‑derived scores—complexity and centrality—jointly explain success probability with Mcfadden's $R^2 = 22.1\%$. This confirms that our synthetic tasks capture an interpretable difficulty gradient, which we can then use to scale difficulty smoothly. This novelty is unavailable without combining synthetic mutation generation with LLM evaluation.
>
> Our analysis on our generated SWE data gives us the following insights:
>
> - **Resource‑scaling diagnostics**. Using the above difficulty axes, we can understand the effect of different resource settings as we scale interaction budget, reasoning gains, etc.. Our statistical tests find that reasoning models are differentially improving at complex functions rather than central ones, and that an increased interaction budget helps with both. The novelty of our problem framing and synthetic testing here allows us to precisely quantify and describe the effect of different resources for LLMs.
> - **Agent behavior analysis**: We add tool usage pattern analysis, revealing that successful models gather significantly more information before attempting repairs and use targeted search effectively. This behavioral analysis provides insights into what makes agents effective at system-level debugging ([see linked figure](https://paste.pics/4f6a27fc7606aea52470be56ff5ab701)).
>
> In short, Breakpoint is reasoning‑diagnostic rather than coverage‑diagnostic: it creates controllable, labelled challenges that let us ask “which resources help where, and why?” This question lies outside the scope of traditional mutation testing and therefore complements rather than competes with it.
>
> Continuing response in next comment.

---

> > ### Comment · Reviewer_zGgD · 2025-06-03
> >
> > Thanks for the very detailed reply, I will use this comment to reply to both of the authors responses
> >
> > >In remove mode, the model is given a function and needs to implement it. This is very similar to classic SWE workflows where given some large codebase a developer needs to implement a well scoped out functionality.
> >
> > Of course this is indeed similar, however, the point here is that the "bugs" or "task" are created synthetically, and I don't see a convincing data-backed argument in the paper that this is realistic compare to what a developer would face in a real-world.
> >
> > >For our discovery mode, this is similar to a bug or backwards compatibility being introduced in a codebase, and then a developer having to diagnose where it lies and fix it. In fact, our evaluation shows much of the difficulty in remove mode lies in finding where the bug is.
> >
> > The author is definitely correct in stating that this is more difficult, since I do think that fault localization -- diagnosing where the bug is is a difficult task, there have been many work that has shown/develoepd tools for fault localization. Again my argument is not that this is not difficult, but that there does not seem to be any data-driven argument that it is realistic
> >
> > My main point here is also not to disparage the arthors on this point of the dataset being realistic or not. I think the difficulty of the dataset is a selling point already (although a small point here may be that if its too difficult would it still be realistic?) and I think the authors should mention very clearly that the dataset is created synthetically
> >
> > >Our goal in this paper is not to develop better mutation testing tools for evaluating software artifacts; instead, it's to apply automated code transformation methods (including, but not limited to, mutation) to understand if and how an LLM‑powered agent can reason about and repair a fault whose effects cascade through a real, multi‑module codebase.
> >
> > I agree that the paper is not just doing mutation testing techniques, but the idea applies here. The point is to transform an previously non-buggy code into a buggy code through a varity of different methods (e.g., mutation generation). There has been a lot of prior work on this topic [1,2,3] (none of which is written by the reviewer). I understand from the authors point of view that this paper deals mainly with how these bug-injection can affect agent's and there are definitely some innovations made
> >
> > [1] HyperPUT: Generating Synthetic Faulty Programs to Challenge Bug-Finding Tools
> > [2] Automated Bug Generation in the era of Large Language Models
> > [3] Learning to construct better mutation faults
> >
> > >Our experiments evaluate five different state-of-the-art models: GPT-4o, o4-mini, Claude-3.7-Sonnet, o3-mini, and Deepseek V3. These represent diverse model families and training approaches (reasoning vs non-reasoning models).
> >
> > I would like to see the authors apply existing state-of-the-art agents, many of which are open-source and can be seen on the swe-bench leaderboard. I feel that this would greatly contribute to the affect of the paper and show to the readers how these popular tools work on the new benchmark
> >
> > > We thank the reviewer for bringing COMMIT0 to our attention. We note that this paper is not in the citation period, but we are happy to add this related work to our paper.
> >
> > Yes the author is right about the paper date, apologize for that. I mainly added this related work as the GitHub repo has been available much earlier https://github.com/commit-0/commit0 (Sept 2024)
> >
> > Overall, I am happy to increase my score if the authors can commit to testing out state-of-the-art existing agent-based approaches in their new revision of the paper (does not need to be done for this review)

---

> > ### Author Response · Authors · 2025-06-05
> > **Further Discussion**
> >
> > We're grateful for your willingness to engage in depth and especially for your openness to increasing your score based on commitments we will make. Below, we respond directly to the main points you've raised.
> >
> >
> > **Realism of Synthetic Tasks:**
> > Thank you for your suggestions: we will explicitly highlight in the paper that Breakpoint’s tasks are synthetic. We agree that the main goal of this benchmarking methology is to allow for scalable difficulty increases even as models continue to get better, not to perfectly mimic the real world. As we mentioned in our reviewer 2 rebuttal, we agree that it could be interesting to add richer, more realistic development signals for the model to use, but thought that was out of the scope of this benchmark as is.
> >
> >  Even though our task is synthetic, our corruption strategies align with empirical bug patterns: function deletion mirrors "missing functionality" bugs that comprise 64-69% of real bug impacts (Tan et al., 2014), while our adversarial corruptions naturally generate semantic errors—the dominant bug type at 81-87% in production systems (Li et al., 2006). The statistical properties of our discovery task also match real-world distributions: median bug fixes affect only 4 lines of code (Sobreira et al., 2018), and our multi-function corruption mode reflects the 20-40% of bugs requiring changes across multiple components (Zhong & Su, 2015). We will add this methodological justification to the appendix.
> >
> > We also wanted to highlight that we've added a preliminary human evaluation section to the paper, which we've written into our review 2 response, where we make sure that the tasks are solvable but not trivial for humans (30 minutes to 4 hours). This doesn't mean that they're copies of real world dev workflows, but does help calibrate what these scores mean for model performance.
> >
> > **Mutation Testing and Novelty:**
> > We acknowledge the overlap with LLM-mutation testing, and we incorporated your cited references ([1,2,3]) into our paper. We appreciate your comment that our novelty lies in both our methodology for testing (graph-aware corruptions).
> >
> > We also wanted to highlight novelty in our analytical methodology, particularly in our differential analysis of what questions performance can be differentiated between models and between resource regimes.
> >
> > **Evaluating SWE-Bench Agents:**
> >
> > We fully commit to evaluating existing open-source state-of-the-art SWE-bench agents in the next revision. The reason we use our custom agent framework is that we wanted a standard comparison framework to fairly evaluate the different models against each other, and many of these agent frameworks are "tuned" to a particular model. However, we agree that using these open source frameworks would be a good comparison point and clarify some of the claims around the benchmark's difficulty.
> >
> >
> > We hope these clarifications underscore both our commitments and the standalone contribution of the benchmark, and we warmly invite any further guidance.

---

> > > ### Comment · Reviewer_zGgD · 2025-06-05
> > >
> > > Thanks for the response again, I hope the authors can apply some of the sota agents to this benchmark and see how they perform. I have increased my score.

---

> ### Author Response · Authors · 2025-06-03
> **Continuing feedback response**
>
> > Evaluation uses only one agent
>
> Our experiments evaluate five different state-of-the-art models: GPT-4o, o4-mini, Claude-3.7-Sonnet, o3-mini, and Deepseek V3. These represent diverse model families and training approaches (reasoning vs non-reasoning models).
>
> Beyond evaluating different models, we conduct extensive ablations on agent configurations:
>
> 1. Tool budget scaling: We vary tool-use iterations (4, 8, 16, 32) and submission attempts (1, 2, 4, 8) to understand how increased computational resources affect performance ([see linked figure](https://paste.pics/0da098285f3ba5998903a17a79b0cd4f)).
> 2. Tool preference analysis: We analyze how different models use available tools, finding that successful models prefer high-level tools like read_file over more granular read_function, and gather significantly more information before attempting repairs ([see linked figure](https://paste.pics/4f6a27fc7606aea52470be56ff5ab701)).
>
>
> > Missed Commit0 related work
>
> We thank the reviewer for bringing COMMIT0 to our attention. We note that this paper is not in the citation period, but we are happy to add this related work to our paper.
>
> While also focused on the same domain of LLM agents solving inverse problems, the cited paper is different in focus than ours in two main ways.
>
> 1. System comprehension vs. greenfield development: COMMIT0 evaluates agents' ability to build libraries from scratch given specifications. Breakpoint evaluates agents' ability to understand and repair existing complex systems they didn't create. This distinction is crucial—real-world software engineering often involves debugging and modifying unfamiliar codebases with intricate dependencies.
> 2. Diagnostic vs. generative challenges: Breakpoint's discovery mode specifically tests fault localization—a critical skill where agents must navigate large codebases to identify problems before fixing them. Our finding that diagnosis is often the primary bottleneck (Section 4.1.3) provides insights not capturable in greenfield development tasks.
>
> In light of these clarifications and improvements, would you consider increasing your score for our paper? If not, could you let us know any additional changes you would like to see in order for this work to be accepted?

---

### Official Review · Reviewer_tkG9 · 2025-05-13

**Rating:** 7
**Confidence:** 3
**Ethics Flag:** 1

**Summary:**

This paper introduces Breakpoint, a novel benchmark designed to test large language models (LLMs) on their systems-level reasoning capabilities within software engineering contexts. Unlike traditional code benchmarks focused on localized tasks, Breakpoint constructs challenging inverse problems by corrupting real-world software repositories—either through adversarial edits or deletions—to require agents to identify and repair the corrupted components using tool-based interactions and test-suite validation. The authors demonstrate that even the strongest current models (e.g., GPT-4o, o3-mini) fail consistently on the most complex multi-function corruption tasks and that performance is tightly linked to both component centrality (how interconnected a function is) and function complexity (cyclomatic complexity, etc.). The benchmark supports discovery and targeted modes and includes a full evaluation framework for public and custom repositories.

**Reasons To Accept:**

The paper addresses an important but under-evaluated capability in LLMs: reasoning across large, interconnected systems—a prerequisite for safe autonomous agents in real-world coding environments.

Corrupting real-world codebases to generate high-fidelity, scalable, and objective benchmarks (backed by test suites) is an elegant way to produce hard tasks while preserving evaluability.

The benchmark explicitly distinguishes between localized and systems-level reasoning, and between repair and discovery modes, enabling nuanced diagnostics of model capabilities.

The paper introduces task difficulty controls based on call graph centrality and function-level complexity, showing that these correlate well with task difficulty and solver success.

The authors rigorously evaluate multiple models and test-time resource regimes (tool use iterations), and include statistical analyses (e.g., Kruskal-Wallis tests) to support findings.

The release of tools for applying Breakpoint to arbitrary repositories (with contamination control) increases its long-term utility for research and safety evaluation.

**Reasons To Reject:**

The current benchmark is restricted to Python repositories. While the authors argue generalizability, this limits immediate application to other domains (e.g., TypeScript, Java, C++) where toolchains, complexity, and error modes differ.

The model operates in a fixed tool interface with limited natural language context or interaction history. In real-world engineering, developers rely on richer signals (e.g., commit messages, stack traces, historical fixes) which are not modeled here.

While success rates and metric correlations are well analyzed, there is little qualitative discussion of why models fail (e.g., hallucinations, tool misuse, misunderstanding of control flow), which could be informative for future model design.

Although the toolkit is advertised as extensible, there is no evaluation of how well model performance or benchmark difficulty generalizes across repositories with varying domain complexity, test coverage, or coding styles.

The paper assumes high centrality/complexity implies difficulty, but doesn’t compare with human performance on these tasks (e.g., how often experienced programmers can repair these corruptions), which would help calibrate difficulty claims.

---

> ### Author Response · Authors · 2025-06-03
> **In response to your encouraging and helpful feedback**
>
> Thank you for your overall supportive review and detailed feedback, which we believe will improve our final manuscript. Please see our responses to the helpful points raised in your review below:
>
> > “Only Python limits immediate application to other domains"
>
> We believe the claims we make are fairly general and should not be substantially different for repositories in other languages.
>
> Our toolkit is easily adaptable to other languages and coding environments, as the only Python specific functionality is in searching for candidate functions by parsing the files. Our code takes in a *test command* to run tests before/after corruptions and this command could just be changed for different language/testing environments.
>
> > "In real-world engineering, developers rely on richer signals (e.g., commit messages, stack traces, historical fixes) which are not modeled here."
>
> We do give the model stack traces as test feedback on the state of the codebase, and the model sees the results of the stack trace when it submits modifications to the code.
>
> We think that giving the models richer commit and history signals would indeed be an interesting extension, and we could give our SWE agent access to git to enable that functionality. Given the nature of breakpoint corruptions, giving the model access to a previous "clean" state of the code would trivialize the task. One way to get around this is to introduce multiple edits at once, some of which introduce bugs and some of which don't. We agree that these kinds of extensions would be valuable to explore, but we wanted to keep the scope of the paper focused on ways to generate tasks by modifying a codebase at one point in time.
>
> However our results remain interesting beyond the scope of trying to mirror SWE workflows. We wanted to develop a method that can straightforwardly generate tasks of nearly arbitrary difficulty. In this case, we think that depriving models of signals that human SWEs have is a good way to make harder problems than mirroring alone will allow.
>
> > Little qualitative failure analysis
>
>
> We thank the reviewer for this feedback. We've added additional analysis on failure modes. For instance, we examine the proportion of the time the model finds the right function in discovery mode ([see figure](https://paste.pics/a51f967d6f4260239899867c9bf3ab51)). In doing so, we find that Claude-3.7-sonnet and o4-mini, which have very similar performance in discovery mode, actually have very different solve profiles -- Claude finds more correct functions, and o4-mini has a much higher solve rate once it finds the right function. In general, the majority of the reason at least for failure of reasoning models in discovery is in the task of finding what function was corrupted.
>
>
> We also look at how much better models get on their additional submission attempts. Here, we find that models rarely improve past their second submission attempt (see linked figures for [discovery](https://paste.pics/d6c7fef1dcfcf7990e1d01e170d98fdc) and [remove](https://paste.pics/799f6b1e96f30ef13b34a81cee8eae64)).
>
>
>
>
> > Analysis of generalizability across diverse repos / test coverage
>
>
> Breakpoint's methodology is inherently adaptable to diverse repositories, regardless of size, coding style, or domain. To ensure our findings generalize, we selected repositories representing a wide variety of software domains, including large parsing libraries, machine learning tooling, database systems, and others.
>
> Moreover, in the revised manuscript, we analyzed whether repository-specific factors influenced task difficulty, including: repository size, test coverage, domain (ML/parsing/databases), and coding style conventions. We found that, while average performance varied substantially between repositories, we found no statistically significant correlations between these repository characteristics and agent performance.

---

> ### Author Response · Authors · 2025-06-03
> **Continuing feedback response**
>
> > Human calibration
>
> **Response.**
>
> We thank the reviewer for this suggestion. We added the following preliminary evaluation appendix section to our paper:
>
> To better contextualize the difficulty of Breakpoint tasks, we conducted a preliminary human evaluation with the authors solving a small set of benchmark tasks. Specifically, four problems from each of the \textsc{remove} and \textbf{discovery} modes were manually attempted. All eight tasks were successfully solved, with completion times ranging from 4 to 30 minutes per task. These results suggest that Breakpoint tasks—at least at lower corruption complexities—are within reasonable human solvability.
>
> We also qualitatively assessed the difficulty of the hardest Breakpoint problems by sampling one hard-set problem for both remove and discovery. Based on our human evaluation, we found that the tasks are quite challenging and would require several hours of human time to understand and fix the codebase, although they are solvable. This supports the human calibration of our claims on difficulty, as our designated hard sets do seem indeed much harder.
>
> To facilitate future systematic comparisons, our benchmark release includes explicit support for human evaluations, providing users with interfaces to efficiently explore, diagnose, and repair tasks directly within the codebase environment.
>
>
> In light of these clarifications and improvements, would you consider increasing your score for our paper?

---

### Official Review · Reviewer_CJEp · 2025-05-17

**Rating:** 7
**Confidence:** 3
**Ethics Flag:** 1

**Summary:**

The paper introduces Breakpoint, a code repair benchmarking strategy for evaluating the abilities of large language models (LLMs) in software engineering tasks. Unlike prior work that focuses on localized, isolated problem-solving, Breakpoint adversarially corrupts real-world GitHub repositories to create challenging repair tasks that require understanding complex dependencies across codebases. It evaluates models in both targeted and discovery modes, using unit tests to verify correctness. Task difficulty is controlled using human-selected metrics like function centrality and cyclomatic complexity, enabling fine-grained analysis of model limitations.

**Questions To Authors:**

The term “Systems-Level Reasoning” is somewhat ambiguous. Could you clarify its definition and provide appropriate citations? the use of this term may be confusing given its potential overlap with concepts like “System 2 reasoning” from cognitive science. A clearer distinction or alignment would be helpful.

"Releasing a complete toolki...", could you please clarify where this release can be found? I couldn't locate any attachment with the submission. Do you plan to open-source your implementation, or has it already been released?

**Reasons To Accept:**

1. The results show that even state-of-the-art models, including GPT-4o and o3-mini, struggle with high-complexity and multi-function corruption scenarios, achieving negligible progress on the hardest tasks.

2. Tackles a critical gap in evaluation, and the benchmark can be re-created or targeted to specific software suites, without being restricted to a particular domain or repository.

3. Proposes a general framework for evaluation in the code repair task.

**Reasons To Reject:**

1. The task difficulties are not entirely convincing. Could you elaborate on why the proposed heuristics sufficiently capture and explain task difficulty?

2. The claim regarding data contamination is unconvincing. If a repository has already been seen during pretraining, generating repair tasks from it does not inherently make the evaluation free from contamination.

3. The evals are mostly guided by test suites. That mean the correctness of this relies on a rigorous test suites automations which is one of the critical but underexplored part of the paper.

---

> ### Author Response · Authors · 2025-06-03
> **Part 1**
>
> We sincerely thank the reviewer for their comments which have been very helpful in improving the paper. We are also grateful for your recognition that our work "tackles a critical gap in evaluation" and provides a benchmark that "can be re-created or targeted to specific software suites, without being restricted to a particular domain or repository." We are also encouraged by your acknowledgment that we "propose a general framework for evaluation in the code repair task".
>
> > "The task difficulties are not entirely convincing. Could you elaborate on why the proposed heuristics sufficiently capture and explain task difficulty?"
>
> We thank the reviewer for this question. We answer the following subquestions.
>
> 1. Why did we expect that these axis capture task difficulty a priori?
>
> Our goal was to differentiate two capabilities:
> - Localized reasoning: fixing a self-contained bug or implementing a single, clearly-defined feature.
> - System-level reasoning: diagnosing and coordinating changes across complex, interconnected codebases.
>
> As a proxy for the first capability, we measure complexity of the corrupted function itself, to quantify the difficulty inherent to *the code the model will have to generate*.
>
> As a proxy for the second capability, we measure call graph centrality, to quantify the difficulty that is inherent to *properties of the codebase outside the function*, through the dependencies that have to be understood to fix the original function.
>
> 2. To what degree were we able to explain task difficulty with our axes?
>
> We find that these two axes explain difficulty fairly well. In remove mode, for instance, the two static measurements of complexity are both correlated with more difficult problems (18.3% and 21.6% McFadden's $R^2$), and jointly they explain problem difficulty better than they do individually by AIC. We find that, by Mcfadden's $R^2$, the joint regression explains $22.1\%$ of the variance, which for McFadden's indicates excellent fit (McFadden 1974).
>
> In our additional figures, we show the results of regressing difficulty onto [function complexity and code-line count](https://paste.pics/ccfad139ba9ad8272cc59525f8211341), and [both together](https://paste.pics/7cdd1c43cdfb8387c66f7c4ec4b33a00). Follow the links for the relevant figures.
>
>
> We also find that problems with high function complexity and centrality are particularly difficult for the model: for problems in the 90th percentile of both these axes, we find that model performances are between 46% and 31% worse percent on remove questions ([see attached figure for detailed result](https://paste.pics/fea9cf1ad43003ed231047a9cc90b548)).
>
> We have reworked the results section substantially to highlight these lines of analysis. We agree that these proxies cannot not fully explain the difficulties via this static decomposition, and that is very unlikely that any particular static decomposition can fully explain task difficult. We mainly claim that these axis have some explanatory power, and that they are sufficient for scaling problem difficulty.
>
> > “The claim regarding data contamination is unconvincing. If a repository has already been seen during pretraining, generating repair tasks from it does not inherently make the evaluation free from contamination.”
>
> We have revised our paper's claims on data contamination: we meant to claim that the specific problem instances contained in Breakpoint are novel, but we agree that the base repositories themselves may have been seen before. In our revision, we highlight the following additional points:
> 1. Our methodology can be directly applied to private code repositories, which means that it can be updated to minimize contamination as time goes on. We will more clearly differentiate this "future-proofing" from claims about contamination.
> 2. Even though the questions are plausibly trained on, frontier models like o4-mini still struggle to solve them.
> 3. This issue is also less significant for the discovery part of our benchmark because we create fully novel corruptions using an LLM. This means the model has to find the source of a previously unseen bug in the code.
>
> > "Test-suite dependence is underexplored"
>
> We agree test suites alone may not cover all failure modes. We do make an effort to filter for high quality corruptions and questions:
> 1. we only evaluate on problems where the corruption causes at least 5 tests to fail, which makes it harder for the model to get the tests to pass without addressing the core functionality
> 2. The repositories we select are popular, with at least 1000 github stars. This means that it's quite likely that the testing is correct, if not complete.
>
> Even if the test suite for a given corruption might not fully capture the function behavior, we believe that this repair task is still a valuable signal of model capability, signaling its ability to diagnose and fix issues specified only by stack traces.

---

> ### Author Response · Authors · 2025-06-03
> **Part 2**
>
> > "Systems-level reasoning" term is ambiguous
>
> We thank the reviewer for highlighting this ambiguity. By 'system-level reasoning,' we refer specifically to the ability to understand and manipulate interconnected components within complex systems, where modifications to one component can cascade through dependencies to affect distant parts of the codebase. This concept draws from systems thinking (Bertalanffy, 1968; Meadows, 2008), which emphasizes understanding wholes through their interconnections rather than isolated parts. We distinguish this from 'System 2 reasoning' (Kahneman, 2011), which refers to deliberate versus intuitive thinking modes—an orthogonal concept, as one could apply System 2 thinking to either local or system-level problems. In our work, we operationalize system-level reasoning through call-graph centrality metrics and multi-function corruption tasks that require understanding how components interact, rather than analyzing them in isolation. We will clarify this distinction by making this definition explicit in the revised manuscript to avoid confusion.
>
> > Toolkit release
>
> We are currently open sourcing our toolkit, and will add the link to the paper in the camera-ready. For anonymization purposes we won't share the link, but could share a link to an anonymous zipfile if it would be useful.
>
> In light of these clarifications and improvements, would you consider increasing your score for our paper? If not, could you let us know any additional changes you would like to see in order to improve this paper further?

---

### Author Response · Authors · 2025-06-03
**General Response to Reviewers**

Dear Reviewers,

We sincerely appreciate your thoughtful and constructive feedback. Your comments have been instrumental in refining our paper. We are encouraged by your positive recognition of Breakpoint’s contributions, particularly:

- Our methodology of adversarially corrupting real-world codebases, allowing for scalable, high-fidelity benchmarking backed by robust test suites.
- Addressing the neglected capability of reasoning across large, interconnected systems, essential for safe and effective deployment of agents.
- Introducing meaningful difficulty metrics (call graph centrality and function-level complexity) that map on to different kinds of reasoning.

Responding directly to your valuable feedback, we have made substantial improvements:

- **Detailed Complexity Analysis**: We deepened our investigation into complexity and centrality metrics, confirming their predictive power and providing clearer insight into their interaction with task performance.
- **Expanded Evaluation**: We broadened our dataset significantly, incorporating more diverse repositories (e.g., parsing libraries, ML tooling, databases), reinforcing the generalizability of our claims.
- **Agent Analysis**: We investigated further on the agent behavior, studying model failures and tool usage to better understand model behavior.


Below, we address your individual points thoroughly and clarify the changes reflected in the revised manuscript. We welcome any additional suggestions to strengthen our contribution further.

Warm regards,

Authors

---

### Author Response · Authors · 2025-06-10
**Request for response**

Dear Reviewers CJEp (R1) & tkG9 (R2),

We’ve now folded all the applicable points point from our earlier exchanges into the revised PDF/rebuttal. If anything still needs clarification, please let us know: happy to respond right away. Otherwise, we’d be grateful if you could register any final comments or score updates before the discussion window closes on 10 Jun AoE.

Thank you again for the thorough feedback and for helping improve the paper!

---

### Decision · Program_Chairs · 2025-07-08

**Decision:**

Accept

**Comment:**

The goal is to ensure we are properly evaluating SWE LLMs, but there are issues of current benchmarks being to narrow/local, lacking applicability to novel or private codebases, and not requiring more complex reasoning over interactions. By corrupting actual code-bases (synthetic introduction of bugs) this work mirrors a more general common practice, but extends it to agents and addresses the three aforementioned concerns. There is some disagreement about the nature of these corruptions in terms of naturalness and connection to the larger literature (both of SWE and existing SWE-Bench agents vs simply the best current LMs).

Despite these points that will all strengthen the work, generally reviewers were positive on the work and recommend acceptance.